# Quantification of potential methane emissions associated with organic matter amendments following oxic soil inundation

Brian Scott[1], Andrew H. Baldwin[1], Stephanie A. Yarwood[1]

[1]Environmental Science and Technology Department, University of Maryland, College Park, 20742, USA

*Correspondence to*: Brian Scott (bscott33@umd.edu)

## Abstract

Methane ($CH_4$) emissions are a potent contributor to global warming and wetlands can be a significant $CH_4$ source. In a microcosm study, we evaluated how the practice of amending soils with organic matter as part of wetland restoration projects may affect $CH_4$ production potential. Organic amendments including hay, manure, biosolids, composted yard waste, and wood mulch were evaluated at three different levels. Using 1-liter glass microcosms, we measured the production of biogenic gases over 60 days in two soils designated by texture: a sandy loam (SL) and a sandy clay loam (SCL). Fresh organic amendments increased $CH_4$ production, leading to potentially higher global warming potential and wetland C loss, and $CH_4$ production was more pronounced in the SL. We observed biogenic gas production in two sequential steady state phases: Phase 1 produced some $CH_4$ but was mostly carbon dioxide ($CO_2$) followed by Phase 2, two to six weeks later, with higher total gas and nearly equal amounts of $CH_4$ and $CO_2$. If this is generally true in soils, it may be appropriate to report $CH_4$ emissions in the context of inundation duration. The $CH_4$ from the SCL soil ranged from $0.003 - 0.8$ cm$^3$ Kg$^{-1}$ day in Phase 1 to $0.75 - 28$ cm$^3$ Kg$^{-1}$ day in Phase 2, and from the SL range from $0.03 - 16$ cm$^3$ Kg$^{-1}$ day in Phase 1 to $1.8 - 64$ cm$^3$ Kg$^{-1}$ day in Phase 2. Adding fresh organic matter

(e.g., hay) increased ferrous iron ($Fe^{2+}$) concentrations whereas in some cases composted
organic matter decreased both $Fe^{2+}$ concentrations and $CH_4$ production. Methanogenesis
normally increases following the depletion of reduceable Fe; however, we observed
instances where this was not the case, suggesting other biogeochemical mechanisms
contributed to the shift in gas production.
**Keywords** Methane emissions, mitigation wetlands, organic amendments
**1 Introduction**
The ecological benefits of wetlands are well documented, including their role as

carbon (C) sinks to stabilize global climate (Mitsch et al., 2015). Driven in part by this
ecological contribution, from 1970 to 2015 human-made wetlands have increased 233%
(Darrah et al., 2019). Between 2004 and 2009 the United States saw a net gain of 16,670
hectares of freshwater wetlands: 360,820 hectares of new wetlands to offset 344,140
hectares of existing (presumably C-sink) wetlands that were destroyed (Dahl, 2011).
Although created or restored wetlands may effectively sequester C, it may take hundreds
of years to offset their radiative forcing due to methane ($CH_4$) emissions (Neubauer,
2014). With such a large number of human-made wetlands, and their potential to increase
global warming, it is vital to consider factors that may contribute to $CH_4$ emissions.

Organic amendments such as straw, wood mulch, manure, and biosolids, mixed

into the soil, are thought to accelerate C storage by enhancing the conversion of plant-
derived compounds to microbial residues (Richardson et al., 2016). Microbial residues,
largely aliphatic-C from cell membrane lipids, can accumulate in soil and are not directly
accessible by methanogens (Chen et al., 2018). Plants contribute both above and
belowground organic matter (OM). Belowground plant materials are preferentially
converted to soil organic carbon (SOC) (Mazzilli et al., 2015). In saturated soils root
residues of wetland plants contain suberin and cutin (Watanabe et al., 2013), which
persist, reducing biogenic gas production (Mikutta et al., 2006). Before contributing to
SOC, standing litter in natural wetlands is partially decomposed by fungi (Kuehn et al.,
2011), and further decomposed by aerobic bacteria (Yarwood, 2018). Allochthonous
organic amendments are derived from above-ground material, but they have not been
subjected to wetland biogeochemical processes. Studies suggest these materials are less
amenable to soil C stabilization compared to natural plant inputs and may increase $CH_4$
production (Scott et al., 2020). In addition to increasing $CH_4$ production directly, organic
amendments may cause SOC priming that produces additional $CH_4$ (Nottingham et al.,
2009), and can lead to an increase in iron (Fe) reduction and toxicity (Saaltink et al.,

2017).

Iron oxides play multiple roles in anoxic soils, being both an electron acceptor for

organic C metabolism (Straub et al., 2001), and a stabilizing agent for SOC on mineral
surfaces (Lehmann and Kleber, 2015). As a metabolite, Fe reduction competes with $CH_4$
production (Huang et al., 2009) and can facilitate sulfur recycling (which also competes
with $CH_4$ production) in freshwater sediments (Hansel et al., 2015). However, recent
literature suggests the relationship of Fe reduction and methanogenesis is more complex.
Some methanogens appear capable of switching between methanogenesis and Fe
reduction (Sivan et al., 2016). In cultures with *Methanosarcina acetivorans*, adding Fe
oxides increased methane production (Ferry, 2020), presumably by the utilization of a
metabolic pathway where electron flow is bifurcated with some electrons going toward
Fe reduction to increase energy yield (Zhuang et al., 2015; Prakash et al., 2019). In
systems that are near pH neutral, Fe reduction does not necessarily have an energetic
competitive advantage over $CH_4$ production (Bethke et al., 2011). In addition to
influencing metabolic pathways, metal-oxide surfaces can stabilize organic matter,
making it less bioavailable, which can affect both Fe reduction (Poggenburg et al., 2018),
C mineralization (Amendola et al., 2018; Lalonde et al., 2012) and production of $CH_4$.

We carried out a lab experiment using organic amendments commonly used in

wetland restoration (biosolids (Bloom®) - B, manure - M, composted yard waste
(LeafGro®) - L, wood chips - W, and hay - H) and measured how they affected $CH_4$
production and Fe reduction.  One-liter (1-L) glass jar microcosms were incubated with
two different soils collected from sites where freshwater wetlands were recently created.
The microcosms were kept under anaerobic conditions to compare the ability of these
substrates to support anaerobic metabolism. We hypothesized that organic amendments
would stimulate dissimilatory Fe-reduction in soils (measured as soluble ferrous iron,
$Fe^{2+}$). Further, we hypothesized that amendments promoting Fe reduction would limit
methanogenesis. We also tested differences between cured (i.e., aged/composted) and
uncured (fresh) organic amendments and hypothesized that uncured amendments would
increase Fe reduction due to the presence of more labile, soluble, compounds. In the
United States organic amendments are often required in mitigation wetlands, that is,
wetlands created or restored to offset wetland losses; however, there has not been a
systematic evaluation of whether or not amendments promote hydric soil conditions (Fe
reduction), lead to Fe toxicity (from Fe reduction), or increase $CH_4$ production.
**2 Materials and methods**
**2.1 Microcosm setup**
Saturated incubations were established using soil from two recent mitigation
wetlands located in Maryland, USA. The first site (76°50'40.35"W, 38°47'5.41"N) was
most recently a horse pasture and will be referred to as SCL denoting the texture (sandy
clay loam). The second site (75°47'40.20"W, 39°1'52.42"N) was most recently a corn/soy
farm with tile drains and was likely a wetland prior to conversion to farmland. The
second site will be referred to as SL (sandy loam). Both sites had been recently graded to
establish wetland topography, so the upper portion of the soils, where soil samples were
collected, were mixed endo- and umbr-aquic horizons but with no ped structure. Soil was
collected from these recently constructed surface horizons to a depth of 15 cm, a typical
depth for mixing-in organic amendments, sieved (2mm) and homogenized prior to use.
Additional soil information is shown on Supplemental Table S1.
Microcosm experiments were conducted in 1-L glass straight-sided wide-mouth
food canning jars. Each microcosm had a total of 600cc of solid material and was filled
with water for a total volume of 660cc. The volumes needed to be precise in order to
facilitate headspace and liquid sampling and to allow space for soil expansion. When
amendments were added, an equal volume of soil needed to be removed so the total
volume of solid material was a constant 600cc. At the start of the experiment, the
headspace was purged with nitrogen gas. The incubation temperature was 20°C. Jar lids
had precision drilled holes fitted with grey butyl rubber stoppers, making it possible to
non-destructively remove the overlying liquid (for Fe and pH analyses) using a 7.5 cm
needle. Since the head-space pressure increased due to biogenic gas production,
atmospheric pressure was re-established during gas sampling events by piercing the septa
with a 24-gauge needle connected to a 50mL gas-tight syringe. This procedure allowed us
to record the total volume of gas produced and collect gas samples (0.01 - 1000 μL)
under atmospheric pressure (Supplemental Figure S1). A small coating of silicone
applied to stoppers after piercing prevented leaks. All microcosm trials were run with
three replicates except where noted.
**2.2 Microcosm Experiments**
**2.2.1 Experiment 1**
We measured $CH_4$ and $Fe^{2+}$ production with various organic amendments,
including composted yard waste (L), composted wood chips (W), class 1 biosolids - (B),
manure (M), and hay (H) at three treatment levels: 8.8% (v/v), 26%, and 53%, in two
soils, a SL and a SCL. We used horse M for the SCL incubations and cow M for the SL
incubations. This matched the wetland mitigation conditions at each field location. The
treatment levels reflect the Maryland Department of Environment (MDE)
recommendation for wetland restoration (60 cubic yards per acre assuming a 6" mixing
depth) = 1x, 3x, and 6x the MDE recommended level. All amendments were sieved to
5mm. Hay was chopped with a Wiley mill, blended, or cut with scissors until it could
easily pass a 5mm sieve.
**2.2.2 Experiment 2**
We measured $CH_4$ and $Fe^{2+}$ production using cured (aged) and uncured (fresh)
organic materials. We used two amendments, B and M. The two cured materials were
from the same two sources as the fresh material but had been cured for a minimum of 3
months. We added the same amount of amendment to each microcosm based on OM
content. Each amendment was evaluated for OM by loss-on-ignition (LOI) (550°C for
2h). Based on the percent OM we adjusted the amount of amendment so the final loading
rate was 20g OM/ 600 $cm^3$ soil. The microcosm setup was the same as Experiment 1
except we used the same volume of soil (600 $cm^3$) in all microcosms. These microcosms
were incubated for 13 days and sampled periodically for $Fe^{2+}$ and biogenic gases.
**2.2.3 Experiment 3**
We measured a) $CH_4$ and b) $Fe^{2+}$ production as a function of pH. We used H
leachate as a substrate (McMahon et al., 2005). We leached 5.63 g H with 125 $cm^3$ cold
de-ionized water, shaking horizontally at 5°C for 24 hours. The leachate was filtered to
20 μm and immediately placed into jars with 600 $cm^3$ SL soil and incubated for 22 days.
The pH was adjusted to target levels of 5.6, 6.1, and 6.6 using a non-substrate buffer: 2-
(N-morpholino) ethanesulfonic acid (MES). To determine the necessary concentration of
MES, we titrated SL (pH 5.8) to our maximum desired pH (6.6). We determined that the
buffering capacity of the soils corresponded to ~ 2 mN in the 125 $cm^3$ of liquid (leachate
volume), so we prepared microcosms using 125 $cm^3$ of 20 mN MES buffer.
**2.2.4 Experiment 4**
We measured $Fe^{2+}$ production using leached H as a substrate (as in Experiment 3)
but compared these finding to those with unleached H, and the H residuals.
**2.3 Soil, Liquid, and Gas Analyses**
Prior to the start of the experiments, we analyzed the SL and SCL for soil texture,
percent soil C, and extractable Fe (Supplemental Table S1). Soil texture was determined
by adding 50 g soil to a 1000 ml cylinder with 0.5% hexametaphosphate. Sand settled
after 1 minute and silt after 24 hours. Soil moisture content was determined as weight
loss of approximately 5 g of soil dried at 105°C for 48 hours. We determined percent soil
C using thermal combustion at 950°C on a LECO CHN-2000 analyzer (LECO Corp., St.
Joseph, MI). Iron extractions were performed sequentially with 1 M hydroxylamine
hydrochloride (HHCL) in 25% v/v acetic acid; 50 g / 1 sodium dithionite in solution 0.35
M ace-tic acid / 0.2 M sodium citrate buffered to pH 4.8; 0.2 M ammonium oxalate / 0.17
M oxalic acid (pH 3.2) (Poulton and Canfield, 2005). The HHCL extraction targets
bioavailable iron, primarily ferrihydrite and lepidocrocite. Dithionite also includes more
crystalline iron oxide forms, hematite and goethite. Oxalate includes the bioavailable iron
oxides and magnetite.
Throughout the experiments we measured $Fe^{2+}$, pH, and biogenic gases in the
headspace. In some cases, $Fe^{2+}$ and pH were measured only at the end of the incubation.
Using a 3" needle, we extracted 0.3 - 1 $cm^3$ (for $Fe^{2+}$) and 1 $cm^3$ (for pH) of the
supernatant liquid to avoid disturbing soil in the jars. Samples of liquid supernatant were
removed during gas sampling, when atmospheric pressure was maintained, to avoid loss
of biogenic gases and atmospheric contamination. For the final sample point the jar
contents were thoroughly mixed prior to sampling to include pore water and gases.
Ferrous iron in supernatant liquid was measured with a HACH DR4000
spectrophotometer. The spectrophotometer was also used to measure Fe in the Fe-oxide
extractions. Prior to analysis, extracted Fe-oxides were reduced by adding thioglycolic
acid. To confirm the spectrophotometer accuracy, a subset of samples was also analyzed
on a PerkinElmer PinAAcle 900T atomic absorption spectrometer. An Orion 9142BN
electrode was used to determine pH.

Gas samples were collected in 12 cm$^3$ N-purged exetainer vials and analyzed by

injecting 5 cm$^3$ into a Varian Model 450-GC gas chromatograph. Since sample volume
was typically 1 cm$^3$ or less, 5 cm$^3$ nitrogen gas was added to the vials immediately prior
to analysis for $CO_2$ and $CH_4$, and measured concentrations were corrected for dilution
and prior headspace gas concentrations. We also performed fluorescent spectral scans on
dissolved organic matter that was extracted from organic materials with 1:10 solid
(weight) / deionized water (volume) for 24 hours and filtered to 0.45 μm (Fischer et al.
2020). After diluting samples, emission spectra were recorded using an Aqualog
fluorometer (Horiba Scientific; Edison, NJ).
**2.4 Data analysis**

Unless otherwise noted, statistical determinations were done using ANOVA in R

or SAS. The $Fe^{2+}$ concentrations were evaluated using contrasts for each of the
amendments compared to the control using the R multcomp package. The gas curves
were modelled as piecewise, bimodal linear functions using the R "Segmented" package
(Muggeo, 2008). Breakpoints were determined using the total gas curves but, in some
cases, Segmented could not identify a breakpoint in the total gas curve, so $CH_4$ curves
were used as noted in Supplemental Figures S2 & S3. Gas curves from H amendments
did not fit a piecewise model and were modelled as sigmoidal functions using the
SSgompertz function in R. However, SSgompertz is sensitive to data scatter, particularly
at the beginning and end of the curve, so the gas curves for H6x in the SL were fitted
with a power function in Excel.
**3 Results**

We present results from four separate experiments, summarized in Table 1. In

Experiment 1, we evaluated Fe and $CH_4$ production by varying OM type and dose, and
soil type (SL vs SCL). In Experiment 2 we controlled other factors and compared
composted versus fresh OM. In Experiment 3 we characterized the effects of pH. In
Experiment 4 we compared iron reduction from the soluble and particulate fraction of
fresh hay, and the results were used to emphasize the pH effect.
**3.1 Experiment 1a: Effect of organic amendments and soil type on $CH_4$ gas production**

Gas production occurred in two distinct steady-state gas production periods,

which we identified as Phase 1, and then after a breakpoint, Phase 2 (Figure 1) with
individual gas curves are shown in Supplemental Figures S2 (SCL) and S3 (SL). Some
$CH_4$ was produced almost immediately upon inundation (Table 2a), but after the
breakpoint (40 days in both the SL and SCL soils), there is a large increase in $CH_4$ as
well as an average 4.7x ± 1.9 increase in total gas production (Table 2b).  One of our
amendments, H, did not fit the linear bimodal pattern, so we reported rates separately on
Table 2c.

Gas production varied by soil texture. In general, the SL soil produced 2.6 times

as much total gas (Figure 2a) and 2.4 times as much $CH_4$ as the SCL (Figure 2b). In the
SCL soil, $CH_4$ production in Phase 1 was 0.003 $cm^3$ $CH_4^{-1}$ Kg soil$^{-1}$ day and with
amendments increased to as much as 0.8 $cm^3$ $CH_4^{-1}$ Kg soil$^{-1}$ day (Table 2a). In Phase 2
1.9 $cm^3$ $CH_4^{-1}$ Kg soil$^{-1}$ day was produced in control soils and with amendments
increased to as much as 28 $cm^3$ $CH_4^{-1}$ Kg soil$^{-1}$ day (Table 2b). In the SL soil,
amendments increased the rate from 0.04 to 16 $cm^3$ $CH_4^{-1}$ Kg soil$^{-1}$ day Phase 1 and from
1.8 to 64 $cm^3$ $CH_4^{-1}$ Kg soil$^{-1}$ day in Phase 2.
Gas production rates generally increased with amendment loading rate (Table 2a
& b), as expected. With the exception of L in the SL, all amendments reduced the time
required to transition from Phase 1 to Phase 2 (i.e. the breakpoint). Biosolids caused the
largest shift, decreasing the breakpoint to as little as 5 days. While amendments generally
increased $CH_4$ production there were exceptions. Low loading rates of cured amendments
(L and W) had lower $CH_4$ production rates than unamended soil: L1 in Phase 1 in both
soils; L3 in the SL; L3 in the SCL (Phase 2 only); and W1 in the SCL (Phase 2).
Biosolids (B1) also lowered $CH_4$ production rates in the SL soil (Phase 1) (Table 2a). We
examined the normalized $CH_4$ production rates (per g C in soil), but in most cases results
were not statistically different at $p < 0.05$ (Supplemental Figure S4). The general trends
indicate uncured amendments (e.g. B and M) produce more methane per unit carbon than
cured amendments (L).
Using fresh H, biogenic gas production followed a sinusoidal pattern and we
reported maximum $CH_4$ production rate at the inflection point (Table 2c). Hay was prone
to floating at higher loading rates and was present in the water column above the surface
(not in contact with soil). In the instances where this occurred (H3 and H6 in the SCL),
there was a decrease in overall gas production rate and very low $CH_4$ – much lower than
unamended soils (Table 2c and Supplemental Figure S2z). Floating also occurred in one
replicated for H6 in SL – the pattern is shown on Supplemental Figures S2&3z, but not
used in the average reported value (Table 2c).
**3.2 Experiment 1b: Effect of organic amendments and soil type on $Fe^{2+}$**
The type and loading rate of organic amendments affected total soluble $Fe^{2+}$
production, compared to the unamended control, in a limited number of cases (Figure 3,
Supplemental Table S2). In the SL soil, L caused a decrease ($p < 0.05$) in supernatant
$Fe^{2+}$ concentrations whereas H increased supernatant $Fe^{2+}$ in both soils ($p < 0.05$). In a
separate set of experiments, we documented the relationship between supernatant Fe and
pore water Fe (Supplemental Figure S5). Soil type affected the amount of soluble $Fe^{2+}$
produced ($p < 0.05$). We did not see a difference in $Fe^{2+}$ in the unamended microcosms
even though the SCL had 2.2x the amount of hydrochloramine hydrochloride extractable
Fe (FeHHCl) compared to the SL and had 7.6x more dithionite extractable Fe
(Supplemental Table S1). Of the FeHHCl in soil, 19% or less in the SCL and 61% or less
in the SL was reduced to $Fe^{2+}$. Hay was an exception, where up to 155 % of the FeHHCl
in the SCL and 236 % in the SL was reduced to $Fe^{2+}$ (Supplemental Table S2). During
the SL soil incubations, aqueous $Fe^{2+}$ was measured simultaneous to $CH_4$ production. In
the H and M treatments, there was a marked increase in $CH_4$ production when $Fe^{2+}$
became asymptotic. However, with the other amendments, $Fe^{2+}$ production continued or
even increased during periods of high $CH_4$ production. Figure 4 shows two examples that
highlight this pattern and the complete set of curves is in Supplemental Figure S6.
**3.3 Experiment 2a: Effect of cured versus fresh organic amendments on CH$_4$ gas production**

In Experiment 1a, it appeared that curing may have had an effect on CH$_4$

production. Fresh H produced the most CH$_4$. The H1 trials had maximum production
rates of 18.2 and 27.8 cm$^3$ CH$_4$$^{-1}$ Kg soil$^{-1}$ day in the SCL and SL soils, respectively
(Table 2c). The H3 and H6 loading rates would likely have been higher had some portion
of the H not floated. The M6 trials produced the most CH$_4$ at 27.7 and 64.0 cm$^3$ CH$_4$$^{-1}$ Kg
soil$^{-1}$ day in the SCL and SL soils, respectively. Of the amendments used, M was cured
the least (after fresh H, which was uncured). LeafGro, a commercial composted yard
waste, was cured the most and produced very little CH$_4$, in some cases less than the
controls. Since we could not specify precisely how long the organic material had been
cured, we conducted a separate experiment with organic materials of known curing
periods (at least 90 days), using B and M. Rather than use the same volumetric quantities,
we used the same loading rate based on OM content. The results confirmed that curing
has a strong influence on CH$_4$ production. Methane production was higher using fresh
material in both cases and cured material sometimes decreased CH$_4$ production (Table 3).
**3.4 Experiment 2b: Effect of cured versus fresh organic amendments on Fe$^{2+}$ production**

In Experiment 1b, we observed that curing also had an effect on the amount of

Fe$^{2+}$ produced. Hay was the only amendment that produced significantly more Fe$^{2+}$ and L
produced a significant reduction in Fe$^{2+}$ (Figure 3). In Experiment 2 we used biosolids
(B) and manure (M) that had been cured at least 3 months. Whether the material had been
cured had a strong influence on Fe$^{2+}$ production and Fe$^{2+}$ was higher using fresh material
in both cases (Figure 5).

**3.4.1 Spectral Analysis: Effect of organic amendments and soil type on CH$_4$ gas production**

      We observed differences in CH$_4$ and Fe reduction rates when using organic material that had been cured versus uncured. The fluorescent spectral signatures of the cured materials (B and M) were similar as were the signatures of fresh material (Supplemental Figure S7), so curing differentiated the materials more than the source. The difference in signatures was indicative of higher concentrations of organic (humic) acids and lower nominal oxidation state in the cured materials. We considered other organic matter characterization methods such as the material's carbon to nitrogen ratio, but we did not find another reliable predictor of CH$_4$ and Fe$^{2+}$ production other than curing.

**3.5 Experiment 3: Effect of pH on a) CH$_4$ and b) Fe$^{2+}$ production**

      The soil pH affected both CH$_4$ and Fe$^{2+}$ production. In Experiment 1, we observed that Fe$^{2+}$ varied with pH in the SL soil (p<0.001; Supplemental Figure S8a), but there was little variation in the SCL (p=0.45; Supplemental Figure S8b). In order to isolate the effect of pH, we performed experiment 3 using a single substrate (H leachate) in the SL soil. Higher pH increased the CH$_4$ production rate in both Phase 1 and 2 (Table 4) and reduced the production of Fe$^{2+}$ (Figure 6).

**3.6 Experiment 4: Leached versus unleached H and pH considerations**

      In Experiment 4 we measured Fe$^{2+}$ produced from H, H leachate, and H residuals (Figure 7). We expected the soluble fraction to be more labile and produce more Fe$^{2+}$; however, the H residuals (solid fraction) appeared to produce more Fe$^{2+}$ than the leachate. As noted on the figure, separate leached fractions changed the system pH. Using the results from Experiment 2, we predict that at comparable pH there would have been

no difference in $Fe^{2+}$ production between H, H residuals, and leachate (Supplemental
Figure S9). Given the potentially strong influence of pH, we re-evaluated the results from
Experiment 2b, correcting for pH and confirmed that the organic material age accounts
for differences in $Fe^{2+}$ production (Supplemental Figure S10). Similarly, we considered
whether pH may have affected the out-come of Experiment 1. A MANOVA analysis of
the Experiment 1 data (Supplemental Table S3) indicated that pH and soil type had a
small effect (p=0.30 and 0.81, respectively) compared to organic matter type and loading
rate (p<0.0001).
**4 Discussion**
Net $CH_4$ emissions are a primary factor that determines whether a wetland is a C
sink or contributes to long term global warming (Neubauer and Verhoeven, 2019). Soil
management practices, such as wetland restoration methods, can have a large impact on
$CH_4$ production and total greenhouse gas emissions (Paustian et al., 2016). Our data
indicate that organic amendments used in created or restored wetlands may have a large
influence on $CH_4$ production. Organic amendments that had been cured (L and W) only
slightly increased $CH_4$ emissions, but fresh material (M and H) resulted in large increases
(Tables 1a&b). This is consistent with field studies where comparable cured amendments
(composted wood and yard waste), did not result in increased $CH_4$ emissions (Winton and
Richardson, 2015), but straw (Ballantine et al., 2015) and peat bales (Green, 2014)
increased $CH_4$ emissions. Organic material is commonly cured, or composted, to remove
plant pathogens (Noble and Roberts, 2004) and to reduce the amount of cellulosic
material (Hubbe et al., 2010), which competes for oxygen, contributing to phytotoxicity
(Saidpullicino et al., 2007; Hu et al., 2011). Curing produces humic acids and increases
the nominal oxidation state (NOSC) of C (Guo et al., 2019). When cured material is then
subjected to anaerobic conditions, less $CH_4$ is produced (Yao and Conrad, 1999), which
would make composted material more suitable in a wetland restoration context.

Following soil inundation, we observed two distinct gas production phases (Phase

1 and 2). This pattern is difficult to distinguish in unamended soils but has been reported
previously (Yao and Conrad, 1999; Drake et al., 2009).  Our breakpoint (5 – 45 days
Table 2b) was similar to Yao and Conrad (1999) (5 – 36 days). The Phase 2 rates in
unamended soils were also similar: $0.96 – 3.98$ $cm^3$ $CH_4^{-1}$ Kg soil$^{-1}$ day in Yao and
Conrad (1999) and $1.82 – 1.94$ $cm^3$ $CH_4^{-1}$ Kg soil$^{-1}$ day in our study (Table 2b).

There are several explanations that could account for the observed gas production

pattern. One is the lag period required to re-establish populations of methanogenic
archaea, which are likely dormant under oxic conditions and regrowth can be on the order
of days (Jabłoński et al., 2015). In our study, B had the earliest shift to Phase 2 $CH_4$
production (Table 2b), possibly due to elevated levels of dormant methanogens present
from anaerobic digestion. The  two-phase gas production could also be due to depletion
of bioavailable Fe-oxides, thus relieving the competition between Fe reducers and
methanogens (Megonigal et al., 2004). Our data were mixed, with some treatments
showing evidence of competition by Fe reducers, but in other cases we did not see
competition. In treatment M1, for example, ferrous Fe in the supernatant plateaued at
about the same time as the breakpoint (Figure 4b), after which $CH_4$ production increased.
In contrast, in W3 soluble Fe continued to be produced well after the breakpoint, and the
amount of bioavailable Fe used during the course of the incubation was less than $28 \pm 4\%$
(Figure 4b, Supplemental Table S2). In addition to quantifying Fe oxide concentrations,
the $CO_2$:$CH_4$ ratios can be indicative of interactions between methanogens and other
reducers (Bridgham et al. 2013). If Fe reduction or other reduction stops during Phase 2,
we would expect the $CO_2$:$CH_4$ ratio to be near 1:1 (Bridgham et al. 2013). However, we
observed notable exceptions. The SCL L1 treatment had a ratio of 73:1 in Phase 2 (Table
2b), yet still had the characteristic shift to higher overall gas production (4.67x). Other
treatments also had higher $CO_2$:$CH_4$ ratios: L3, L6, W1, B1, C, and W1-3 in the SL soil
(Table 2b). Our mixed observations may have been due to microsite formation. In high
producing microcosms, microsite development may have been disrupted by gas
ebullition, which was substantial enough in H amended trials to cause effervescence.
Amendments with low gas production and limited gas ebullition (e.g. L, W and C)
continued to produce $Fe^{2+}$ after the breakpoint, possibly because methanogens were
active in undisturbed microsites, as described in Yang et al. (2017).

The increased gas production from organic amendments was more pronounced in

SL compared to SCL, where there was 2.4x higher $CH_4$ and 2.6x higher gas production
(Figures 2a & b). We observed a more pronounced effect than a recent rice field study
where there was more $CH_4$ from SL soils versus SCL, although in that study results were
not statistically significant (Kim et al., 2018). Yagi and Minami (1990) observed that
compost (approximate loading rate the same as our 1x treatment) increased respiration
rates by 1.8x in a SCL versus a loam soil. Maietta, Hondula, et al. (2020) observed that
respiration rates were higher in a sandy loam soil compared to a silty clay, with and
without 3.3% & 23% wetland hay amendments. Thus, we might conclude that in general
coarser grained (sandy) soil textures emit more $CH_4$; however, there are a number of
investigations where this was not the case (Yagi and Minami, 1990; Glissmann and
Conrad, 2002). Other factors may have contributed. In our experiment the SCL had 7.6x
dithionite extractable Fe, and 4.6x as much %C (Supplemental Table S1), so additional
studies would be needed to isolate texture as the controlling factor.
We considered the gas production from H microcosms separately because they
followed a different pattern than the other amendments, but the pattern was similar to
other studies using hay (Glissmann and Conrad, 2002) and wetland hay (Maietta et al.,
2020b). Our study adds to these findings by observing that H produced very low $CH_4$ in
the water column (after floating) compared to being mixed with soil (Table 2c). This may
merit further study because if this is generally true, applying fresh organic matter as a
mulch, rather than mixed into the soil, could greatly reduce the adverse consequence of
increased $CH_4$ emissions.
Reduction of Fe-oxides occurs in saturated soils in the presence of an organic
substrate and is a key biogeochemical process in wetland soils. With sufficient time,
hydric soils may develop redoximorphic features from Fe reduction; however, studies
have not shown lasting redoximorphic development due to organic amendments (Gray,
2010; Ott et al., 2020). Organizations responsible for constructing mitigation wetlands
have an interest in documenting Fe reduction prior to redoximorphic feature development
as evidence soils that are hydric. Some mitigation wetland practitioners experience
challenges meeting hydric soil testing standards. Although reports in the scientific

literature are rare, there are examples of sites meeting vegetation and hydrology wetland indicators, but not hydric soils (Berkowitz et al., 2014). Both the soils we tested produced sufficient $Fe^{2+}$ and would have passed hydric soils tests, so a soil amendment would not be needed.

We observed that fresh organic matter resulted in increased $Fe^{2+}$ compared to cured organic matter (Figure 3), likely due to the presence of labile carbon, allowing access to more crystalline Fe-oxides (Lentini et al., 2012). In some soils, Fe-reducing bacteria using fresh organic matter amendments could access crystalline Fe making it more bioavailable. However, without an anoxic/oxic cycle, increased $Fe^{2+}$ production could lead to $Fe^{2+}$ toxicity and ferrolysis (Kirk, 2004), similar to the way fresh organic matter leads to SOC priming (Blagodatsky et al., 2010). Ferrolysis occurs when bioavailable Fe-oxides are reduced to $Fe^{2+}$ and are subject to hydraulic transport. We observed that cured amendments, like L, lowered $Fe^{2+}$ concentrations (Figure 3), possibly due to the presence of humic acids that are generated during curing (Guo et al., 2019). Humic acids often contain insufficient biogeochemical energy to drive dissimilatory Fe reduction (Keiluweit et al., 2017), chelate $Fe^{2+}$, removing it from the liquid phase (Catrouillet et al., 2014), and create insoluble precipitates (Shimizu et al., 2013).

Regulating $Fe^{2+}$ production, through the selection of the appropriate OM amendment, could influence the growth of wetland plants. For example, rice growth may be stimulated under low $Fe^{2+}$ doses of 1 mg/L (Müller et al., 2015), but higher doses can produce detrimental Fe plaque (Pereira et al., 2014). Some native wetland species are adapted to high $Fe^{2+}$ concentrations. *Juncus effusus* growth is stimulated at 25 mg/L $Fe^{2+}$

(Deng et al., 2009). North American native reed *Phragmites australis* ssp. *americanus*
was stimulated at 11 mg/L $Fe^{2+}$ from ferrous sulfate (Willson et al., 2017), but the
invasive Eurasian lineage of *Phragmites australis* seedling growth was inhibited by $Fe^{2+}$
as low as 1 mg/L (Batty, 2003). Soils high in free $Fe^{2+}$ adversely affected *P. australis*
growth by creating an Fe-oxide plaque on roots (Saaltink et al., 2017).

Our results show that pH has a significant effect on both the production of $Fe^{2+}$

(Figure 3) and $CH_4$ (Table 3). Between pH 5.6 and 6.6, the lower pH produced more $Fe^{2+}$
and less $CH_4$, consistent with thermodynamic predictions (Ye et al., 2012).
Hydrogenotrophic methanogens can maximize $CH_4$ production at pH 5 (Bräuer et al.,
2004). In rice paddy soils, $CH_4$ emissions had a clear peak at pH 7, but almost none
below pH 5.5 (Wang et al., 1993). The strong effect of pH underscores the need to take
this parameter into account when interpreting data from experiments evaluating Fe-
reduction and methanogenesis. Attempting to control the pH of soils could potentially
introduce confounding effects. We used an MES buffer with 10x the quantity we
estimated from a soil titration and still saw shifts in the pH after incubation. With a high
residual soil acidity, the amount of buffer needed to control soil pH may increase the
ionic strength to a level that could influence cellular sorption to mineral and Fe-oxide
surfaces (Mills et al., 1994) as well as enzyme activity (Leprince and Quiquampoix,

1996).

**5 Implications**

In our experiment, we observed that organic amendments can increase $CH_4$

production, particularly after extended anaerobic periods. We quantified $CH_4$ production
potential from several organic amendments, and in a separate field experiment
(unpublished) show that these results are useful in predicting field $CH_4$ production. There
is mounting concern that $CH_4$ from restored and created wetlands may result in net global
warming for decades to centuries (Neubauer, 2014). Our results suggest that not only do
organic amendments increase $CH_4$ gas production overall, but uncured amendments can
also decrease the time it takes before there is a large increase in both total gas production
and $CH_4$. Methane production is not constant and dramatically increases after several
weeks. Because of this, it may be beneficial to report wetland $CH_4$ data along with
inundation duration, which can strongly affect $CH_4$ (Hondula et al., 2021). It may be
possible to limit $CH_4$ in many wetland settings, particularly mitigation wetlands where
hydrology is part of the design: shorter flooding or inundation durations with alternating
drier conditions. This strategy has been proposed for rice paddy fields (Souza, 2021). Our
lab study demonstrates the potential for significant $CH_4$ emissions, but in a real system,
methanotrophic activity could attenuate some of the emissions (Chowdhury and Dick,
2013); however, this would not decrease the overall C loss from soils, it only changes the
pathway. If organic amendments are to be used, cured amendments may be preferrable
because they are not as prone to high $CH_4$ generation and may attenuate $Fe^{2+}$ toxicity.
Amendments that lower the soil pH increase Fe reduction and limit methanogenesis
(Marquart et al., 2019). When deciding whether or not to use organic amendments for
wetland mitigation consideration should be given to whether or not the material has been
cured, the pH, the soil texture, and expected hydroperiod.

**Figures**

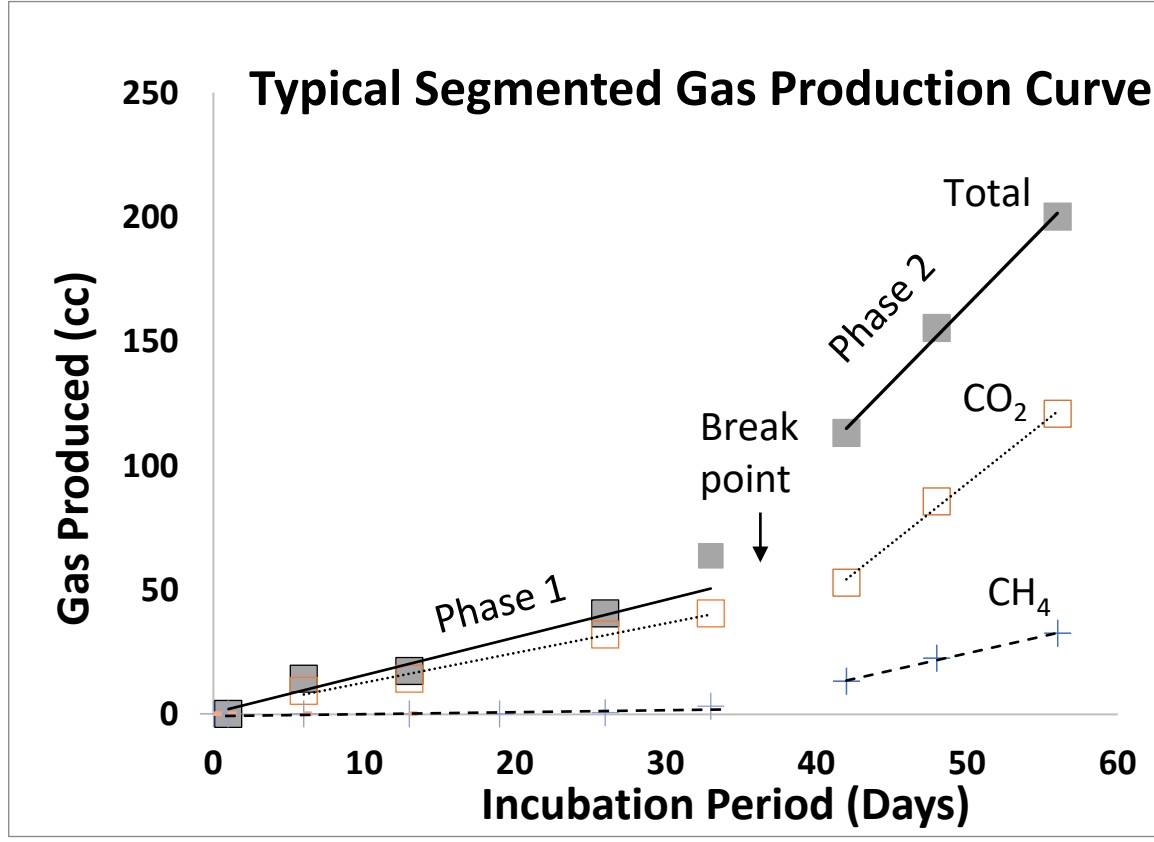

Fig. 1 – Typical gas production in saturated soils amended with organic matter (All
Experiments).
Gases were best modeled using a segmented linear function. After a breakpoint the
average total gas production increased by a factor of ~5 whereas there is a sharp (>> 5x)
increase in methane production. Data presented is from the manure (1x) amended trials in
sandy clay loam soil. Note that hay amended trials exhibited a typical sinusoidal pattern
shown in Supplemental Figures S2&3h,i,j).

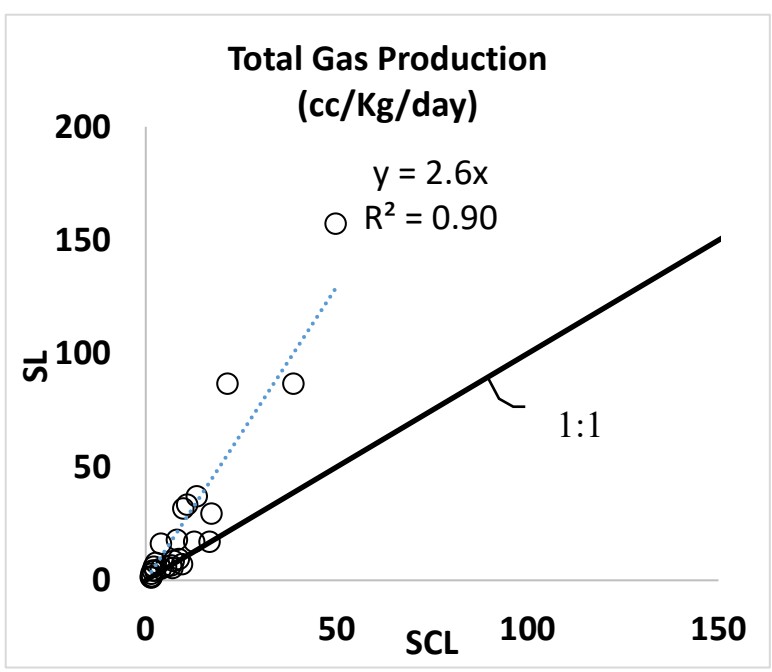

Fig. 2a. Experiment 1. Total biogenic gas production rate in the SL soil versus the SCL mesocosms. The
SL mesocosms had, on average, 2.6 times higher gas production than the SCL.

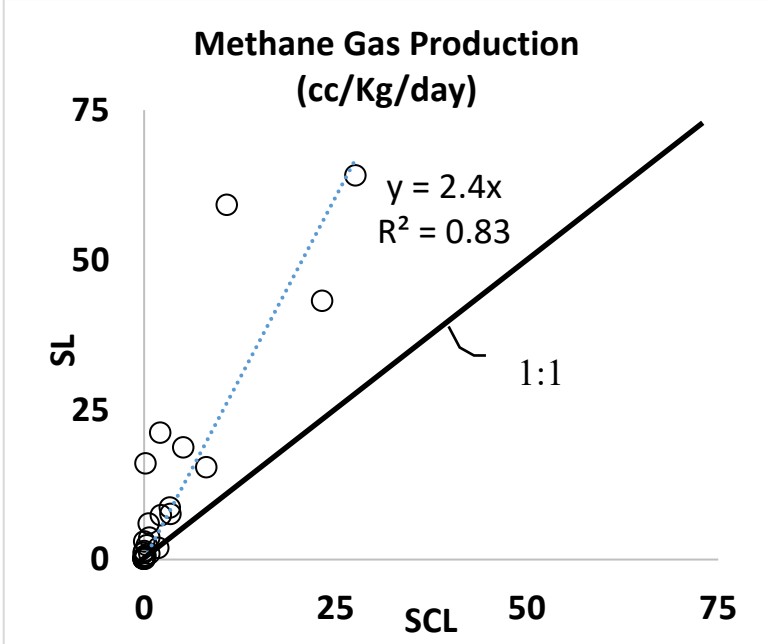

Fig. 2b. Experiment 1. Biogenic methane gas production rate in the SL soil versus the SCL mesocosms.
The SL mesocosms had, on average, 2.4 times higher methane gas production than the SCL.


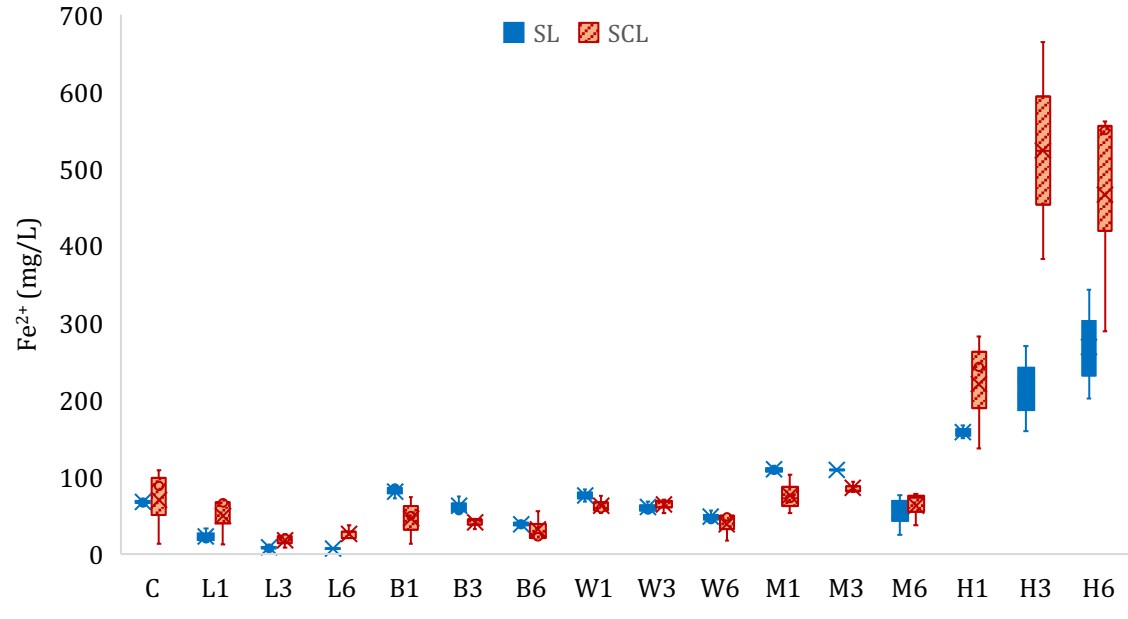


Fig. 3 – (Experiment 1b) Ferrous iron ($Fe^{2+}$) concentration in the liquid phase at the end
of the incubation period.
Microcosms receiving different organic amendment types and levels in Sandy Clay Loam
(SCL) and Sandy Loam (SL) soils. C = no amendment control, L = LeafGro (yard waste),
B = biosolids, W = wood chips, M = manure, H = hay. Numbers signify treatment level
(1, 3, or 6 times amount of organic matter equivalent to 60 $yd^3$ / acre to a depth of 6
inches). Different lower-case letters signify differences (p < 0.05) based on contrasts
compared to C and brackets signify all results in the bracketed group were not
statistically different. H increased total $Fe^{2+}$ production compared to the C in both soils,
and L decreased total $Fe^{2+}$ production compared to C (SL only).

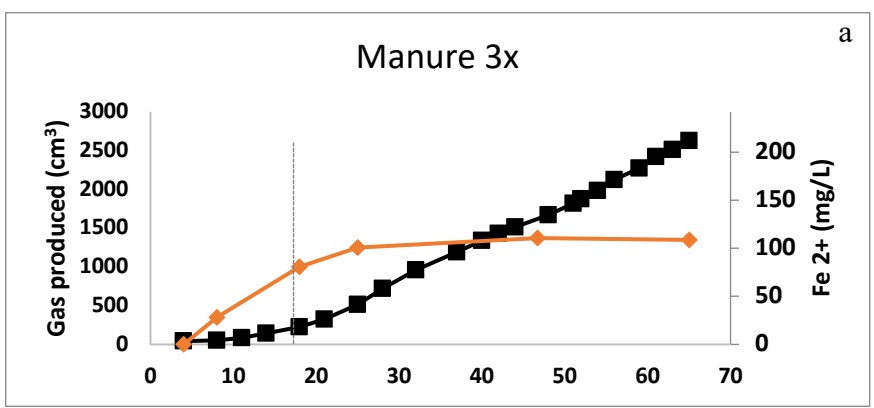

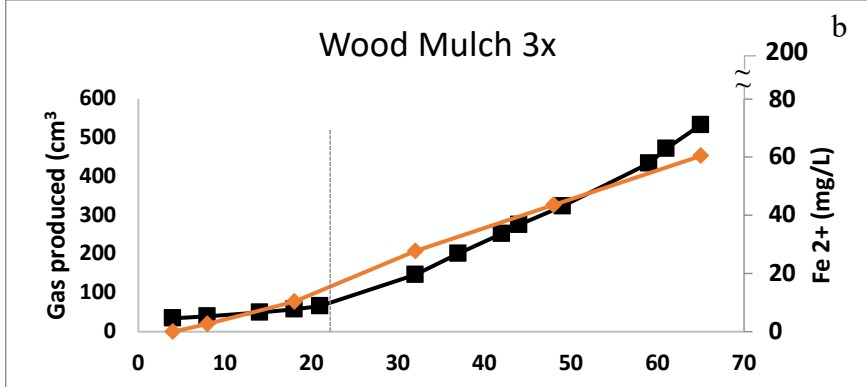

Fig. 4 – (Experiment 1b) Ferrous iron (Fe$^{2+}$) and methane (CH$_4$) in selected microcosms. Depletion of Fe coincided with the breakpoint (dashed line) with manure, but not with wood mulch. Other examples of this pattern are shown in Supplemental Figure S6. The maximum value on the secondary x-axis is the maximum expected Fe$^{2+}$ concentration based on the HHCL extraction.

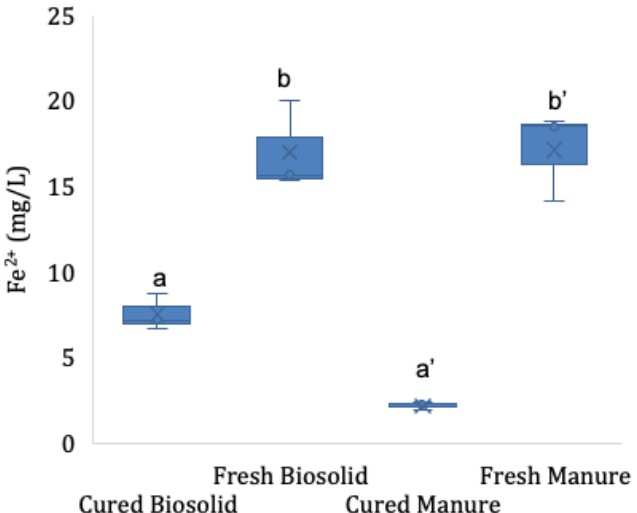



Fig. 5 – (Experiment 2b) Ferrous iron ($Fe^{2+}$) concentration in the liquid phase at the end
of the incubation period (13 days).
Incubation was carried out in sandy loam soil. Different letters indicate a difference at
$p < 0.001$.

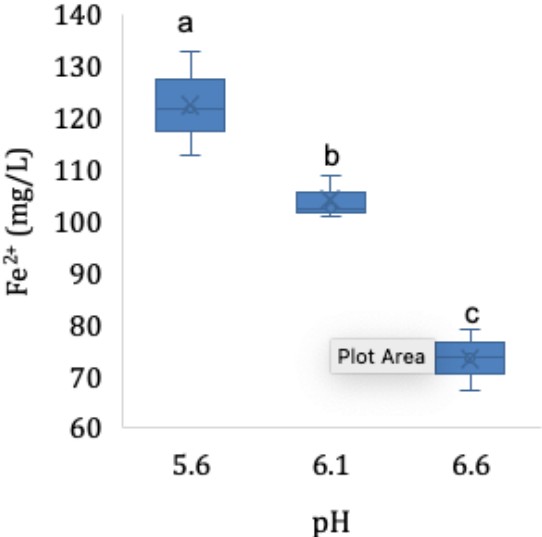

Fig. 6 – (Experiment 3) Ferrous iron ($Fe^{2+}$) concentration in the liquid phase with varied
pH in microcosms receiving hay in Sandy Loam soils.
Different letters indicate a difference at $p < 0.05$.


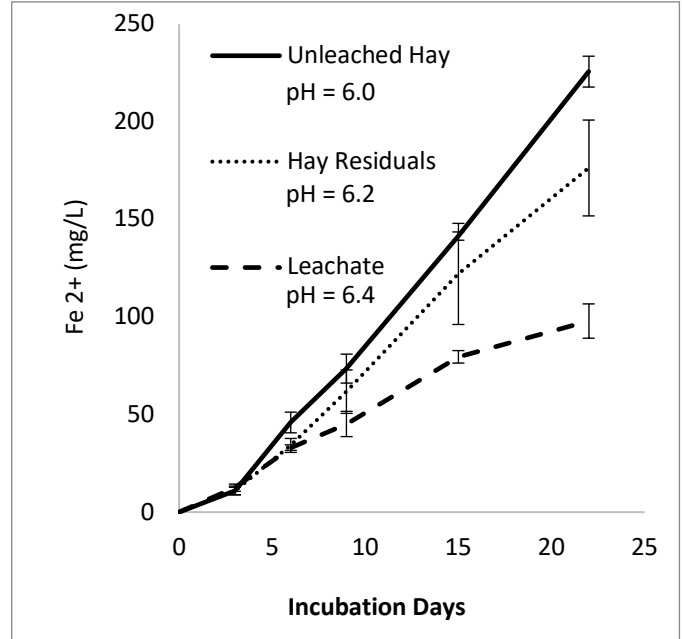




Fig. 7 – (Experiment 4) Ferrous iron ($Fe^{2+}$) concentration in the liquid phase with hay as
substrate.


# Tables


Table 1 – Summary of Results

| Treatment | Effect | | |
|---|---|---|---|
| | Iron Reduction | Methane | Breakpoint |
| Organic Matter | ↔ | ↑ | ↓ |
| Increased Dose | ↘ | ↑ | ↓ |
| Composting/Curing | ↓ | ↓ | ↑ |
| Decreased pH | ↑ | ↓ | N/A |
| SL vs SCL | ↓ | ↑ | N/A |
| Soluble vs. particulate OM | ↔ | N/A | N/A |

538       Breakpoint = time to increased methane production

539       SL = Sandy Loam, SCL = Sandy Clay Loam

↔ No change, ↑ increase, ↓ decrease, ↘ slight decreasing trend
Table 2a – (Experiment 1a – Phase 1). Carbon dioxide ($CO_2$), methane ($CH_4$) and total
gas production. Organic amendment types: B (biosolids), M (manure), L (composted yard
waste), W (composted wood chips) and levels (1 = 60 $yd^3$ / acre equivalent: 3 = 180; 6 =
360) in silty clay loam (SCL) and sandy loam (SL) soils. Instances where organic
amendments did not increase $CH_4$ production are bolded. Note: $CO_2$ : $CH_4$ ratios are
based on calculated gas production rates, not total gas produced.

| Soil | Treatment | Soil (g) | $CO_2$ | | $CH_4$ | | Total Gas | | $CO_2$:$CH_4$ |
|---|---|---|---|---|---|---|---|---|---|
| | | | $cm^3$/day | $cm^3$/Kg/day | $cm^3$/day | $cm^3$/Kg/day | $cm^3$/day | $cm^3$/Kg/day | |
| SCL | Control | 621.63 | 0.97 | 1.56 | 0.002 | 0.003 | 0.99 | 1.59 | 520.0 |
| SCL | B1 | 425.24 | 1.53 | 3.61 | 0.08 | 0.18 | 4.13 | 9.70 | 20.1 |
| SCL | B3 | 544.53 | 1.50 | 2.76 | 0.44 | 0.80 | 3.85 | 7.06 | 3.5 |
| SCL | B6 | 468.02 | 2.09 | 4.46 | 0.06 | 0.13 | 3.53 | 7.55 | 34.3 |
| SCL | M1 | 583.40 | 0.74 | 1.27 | 0.02 | 0.04 | 1.33 | 2.27 | 31.8 |
| SCL | M3 | 495.56 | 1.79 | 3.61 | 0.32 | 0.64 | 2.05 | 4.13 | 5.6 |
| SCL | M6 | 394.39 | 1.49 | 3.77 | 0.12 | 0.30 | 4.35 | 11.03 | 12.6 |
| SCL | L1 | 586.46 | 0.83 | 1.42 | 0.001 | **0.001** | 0.85 | 1.45 | 1420.0 |
| SCL | L3 | 516.34 | 0.89 | 1.72 | 0.01 | 0.01 | 0.91 | 1.77 | 172.0 |
| SCL | L6 | 410.17 | 0.67 | 1.63 | 0.04 | 0.09 | 0.80 | 1.95 | 18.1 |
| SCL | W1 | 593.36 | 1.00 | 1.68 | 0.01 | 0.01 | 0.92 | 1.56 | 168.0 |
| SCL | W3 | 539.61 | 0.98 | 1.81 | 0.10 | 0.19 | 1.39 | 2.58 | 9.5 |
| SCL | W6 | 457.42 | 1.03 | 2.25 | 0.11 | 0.24 | 1.29 | 2.81 | 9.4 |
| SL | Control | 634.60 | 0.50 | 0.79 | 0.03 | 0.04 | 0.56 | 0.88 | 19.8 |
| SL | B1 | 606.80 | 1.25 | 2.06 | 0.02 | **0.04** | 4.13 | 6.80 | 51.5 |
| SL | B3 | 551.50 | 1.57 | 2.84 | 0.44 | 0.79 | 2.92 | 5.29 | 3.6 |
| SL | B6 | 467.87 | 2.08 | 4.44 | 0.59 | 1.27 | 3.81 | 8.15 | 3.5 |
| SL | M1 | 619.92 | 2.62 | 4.22 | 0.58 | 0.93 | 3.49 | 5.63 | 4.5 |
| SL | M3 | 588.37 | 4.48 | 7.61 | 3.44 | 5.85 | 9.42 | 16.02 | 1.3 |
| SL | M6 | 540.93 | 8.63 | 15.95 | 8.59 | 15.87 | 17.92 | 33.13 | 1.0 |
| SL | L1 | 600.10 | 0.35 | 0.58 | 0.02 | **0.03** | 0.73 | 1.22 | 19.3 |
| SL | L3 | 530.30 | 0.61 | 1.15 | 0.02 | **0.03** | 0.78 | 1.46 | 38.3 |
| SL | L6 | 425.87 | 0.62 | 1.47 | 0.11 | 0.26 | 1.66 | 3.89 | 5.7 |
| SL | W1 | 603.27 | 0.98 | 1.62 | 0.06 | 0.10 | 1.55 | 2.56 | 16.2 |
| SL | W3 | 538.77 | 1.42 | 2.64 | 0.20 | 0.36 | 2.14 | 3.98 | 7.3 |
| SL | W6 | 442.57 | 3.05 | 6.88 | 0.24 | 0.54 | 3.23 | 7.31 | 12.7 |


Table 2b – (Experiment 1a – Phase 2). Carbon dioxide ($CO_2$), methane ($CH_4$) and total gas production, and the Phase 1 : Phase 2 breakpoint. Organic amendment types: B (biosolids), M (manure), L (composted yard waste), W (composted wood chips) and levels (1 = 60 $yd^3$ / acre equivalent: 3 = 180; 6 = 360) in silty clay loam (SCL) and sandy loam (SL) soils. Instances where organic amendments did not increase $CH_4$ production are bolded.  Note: r^2 values represent the combined best fit curve, using triplicate samples, for Phase 1 (Table 1a) and Phase 2.

| Soil | Treatment | $CO_2$ | | $CH_4$ | | Total Gas | | $CO_2:CH_4$ | Break Point | r^2 | Ph 2: Ph1 |
| | | $cm^3$/day | $cm^3$/Kg/day | $cm^3$/day | $cm^3$/Kg/day | $cm^3$/day | $cm^3$/Kg/day | | | | |
| --- | --- | --- | --- | --- | --- | --- | --- | --- | --- | --- | --- |
| SCL | Control | 2.06 | 3.31 | 1.20 | 1.94 | 2.54 | 4.09 | 1.7 | 40.0 ± 4.5 | 0.959 | 2.57 |
| SCL | B1 | 5.58 | 13.13 | 1.47 | 3.45 | 5.49 | 12.91 | 3.8 | 29.3 ± 1.9 | 0.987 | 1.33 |
| SCL | B3 | 3.74 | 6.86 | 4.45 | 8.17 | 9.48 | 17.40 | 0.8 | 20.1 ± 3.4 | 0.974 | 2.46 |
| SCL | B6 | 7.42 | 15.85 | 10.90 | 23.29 | 18.20 | 38.89 | 0.7 | 10.3 ± 2.4 | 0.994 | 5.15 |
| SCL | M1 | 2.26 | 3.88 | 1.29 | 2.22 | 5.82 | 9.97 | 1.7 | 40.2 ± 2.1 | 0.997 | 4.39 |
| SCL | M3 | 4.64 | 9.37 | 5.39 | 10.89 | 10.69 | 21.58 | 0.9 | 20.8 ± 0.8 | 0.997 | 5.23 |
| SCL | M6 | 5.85 | 14.83 | 10.91 | 27.67 | 19.69 | 49.93 | 0.5 | 22.1 ± 3.2 | 0.956 | 4.53 |
| SCL | L1 | 3.85 | 6.57 | 0.05 | **0.090** | 3.96 | 6.76 | 73.0 | 32.2 ± 1.6 | 0.966 | 4.67 |
| SCL | L3 | 4.21 | 8.16 | 0.39 | **0.75** | 4.54 | 8.79 | 10.9 | 32.0 ± 2.2 | 0.983 | 4.97 |
| SCL | L6 | 5.90 | 14.39 | 0.92 | 2.24 | 6.95 | 16.95 | 6.4 | 32.0 ± 3.7 | 0.923 | 8.68 |
| SCL | W1 | 1.56 | 2.63 | 0.27 | **0.460** | 3.22 | 5.42 | 5.7 | 34.0 ± 3.7 | 0.986 | 3.48 |
| SCL | W3 | 1.93 | 3.58 | 1.90 | 3.52 | 4.51 | 8.35 | 1.0 | 24.2 ± 3.1 | 0.989 | 3.23 |
| SCL | W6 | 2.19 | 4.79 | 2.36 | 5.15 | 6.22 | 13.60 | 0.9 | 13.0 ± 2.4 | 0.981 | 4.84 |
| SL | Control | 1.00 | 1.58 | 1.16 | 1.82 | 3.11 | 4.91 | 0.9 | 40.0 ± 3.2 | 0.957 | 5.55 |
| SL | B1 | 4.44 | 7.31 | 5.16 | 8.50 | 10.19 | 16.79 | 0.9 | 8.6 ± 3.0 | 0.880 | 2.47 |
| SL | B3 | 8.76 | 15.89 | 8.42 | 15.28 | 16.12 | 29.23 | 1.0 | 4.7 ± 1.8 | 0.989 | 5.53 |
| SL | B6 | 12.61 | 26.96 | 20.15 | 43.07 | 40.39 | 86.33 | 0.6 | 9.1 ± 1.2 | 0.992 | 10.59 |
| SL | M1 | 8.64 | 13.93 | 13.03 | 21.02 | 19.41 | 31.30 | 0.7 | 16.7 ± 0.7 | 0.998 | 5.56 |
| SL | M3 | 15.23 | 25.88 | 34.77 | 59.10 | 50.79 | 86.33 | 0.4 | 17.2 ± 1.5 | 0.992 | 5.39 |
| SL | M6 | 29.50 | 54.53 | 34.62 | 64.00 | 84.92 | 156.98 | 0.9 | 29.4 ± 1.4 | 0.974 | 4.74 |
| SL | L1 | 1.35 | 2.24 | 1.71 | 2.85 | 3.76 | 6.26 | 0.8 | 38.3 ± 1.2 | 0.992 | 5.12 |
| SL | L3 | 2.27 | 4.27 | 1.86 | 3.50 | 4.82 | 9.09 | 1.2 | 40.5 ± 2.0 | 0.977 | 6.22 |
| SL | L6 | 4.25 | 9.99 | 3.07 | 7.21 | 7.15 | 16.78 | 1.4 | 44.8 ± 1.3 | 0.988 | 4.31 |
| SL | W1 | 2.10 | 3.48 | 1.32 | 2.19 | 3.47 | 5.76 | 1.6 | 25.6 ± 7.6 | 0.762 | 2.25 |
| SL | W3 | 6.58 | 12.22 | 4.05 | 7.51 | 9.46 | 17.56 | 1.6 | 23.2 ± 2.3 | 0.974 | 4.41 |
| SL | W6 | 10.10 | 22.83 | 8.23 | 18.60 | 16.22 | 36.65 | 1.2 | 23.2 ± 1.1 | 0.991 | 5.02 |
| | | | | | | | | | | AVERAGE | 4.7 |
| | | | | | | | | | | STDEV | 1.9 |

Table 2c – Experiment 1a. Carbon dioxide ($CO_2$), methane ($CH_4$) and total gas production with hay (H) amendment. H amended trials fit a sigmoidal, not segmented, pattern, and therefore there was no breakpoint and we present p values for the sigmoidal fit, except H6 SL rates where we used a power function in Excel and report the $r^2$ value. Gas production rates (cm 3 gas Kg soil -1 day -1) represent maximum at the inflection point. The amendment floated to the surface in the SCL H3 and H6 trials, which resulted in unusually low $CH_4$ production rates.

| Sigmoidal curve values | | | $CO_2$ | | | $CH_4$ | | | Total Gas | | | |
|---|---|---|---|---|---|---|---|---|---|---|---|---|
| Soil | Treatment | Soil (g) | $cm^3$/day | $cm^3$/Kg/day | p | $cm^3$/day | $cm^3$/Kg/day | p | $cm^3$/day | $cm^3$/Kg/day | p | $CO_2$:$CH_4$ |
| SCL | H1 | 573.03 | 9.70 | 16.93 | 2.0E-16 | 10.40 | 18.15 | 0.164 | 37.1 | 64.75 | 1.3E-12 | 0.93 |
| SCL | H3 | 477.85 | 7.50 | 15.70 | 3.0E-14 | 0.02 | 0.04 | 0.933 | 9.90 | 20.72 | 7.8E-6 | 393 |
| SCL | H6 | 334.20 | 6.60 | 19.75 | 0.019 | 0.09 | 0.27 | 0.921 | 6.70 | 20.05 | 9.6E-13 | 73 |
| SL | H1 | 582.57 | 8.90 | 15.28 | 5.5E-14 | 16.20 | 27.81 | 0.283 | 18.40 | 31.58 | 2.9E-4 | 0.55 |
| SL | H3 | 478.00 | 20.80 | 43.51 | 1.8E-13 | 12.20 | 25.52 | 0.636 | 36.80 | 76.99 | 0.0093 | 1.7 |
| SL | H6 | 321.13 | 50.71 | 158.0 | 0.93(r^2) | 77.7 | 242.1 | 0.69(r^2) | 79.79 | 248.47 | 0.74(r^2) | 0.65 |

Table 3 – (Experiment 2a). Methane gas data for incubations with fresh and cured organic matter in sandy loam soil.

Control data (*) from Experiment 1a (Table 2a) included for reference. Different letters indicate a difference at $p<0.001$.

| Treatment | Phase 1 Methane ($cm^3$/Kg/day) | Phase 2 Methane ($cm^3$/Kg/day) |
|---|---|---|
| Control* | 0.04 | 1.8 |
| Cured Biosolids[a] | 0.003 | 0.37 |
| Fresh Biosolids[b] | 3.29 | 17.48 |
| Cured Manure[a'] | 0.22 | 5.4 |
| Fresh Manure[b'] | 3.85 | 42.36 |

Table 4 – (Experiment 3). Methane gas data versus pH.

Microcosms receiving hay in Sandy Loam soils (Experiment 3). Different letters indicate a difference at $p<0.001$.

| pH | Phase 1 CH$_4$ (cm$^3$/Kg/day) | Phase 2 CH$_4$ (cm$^3$/Kg/day) |
|---|---|---|
| 5.6[a] | 0.44 | 10.6 |
| 6.1[b] | 1.0 | 13.0 |
| 6.6[c] | 1.8 | 13.8 |

**Declarations**
**Funding**
Work was made possible by funding from the Maryland State Highway Administration

(SHA/UM/4-53), the Maryland Water Resources Research Center (2017MD340B), and

USDA National Institute of Food and Agriculture, Hatch Project Number: MD-ENST-

7741.

**Conflicts of interest/Competing interest**
Authors declare no conflict of interest

**Availability of data and material**
Significant data detail is available in the supplementary materials. Additional raw data

available upon request.

**Code availability**
None

Amendola, D., Mutema, M., Rosolen, V., and Chaplot, V.: Soil hydromorphy and soil carbon: A global data analysis, 324, 9–17, https://doi.org/10.1016/j.geoderma.2018.03.005, 2018.

Ballantine, K. A., Lehmann, J., Schneider, R. L., and Groffman, P. M.: Trade-offs between soil-based functions in wetlands restored with soil amendments of differing lability, 25, 215–225, https://doi.org/10.1890/13-1409.1, 2015.

Batty, L. C.: Effects of External Iron Concentration upon Seedling Growth and Uptake of Fe and Phosphate by the Common Reed, Phragmites australis (Cav.) Trin ex. Steudel, Annals of Botany, 92, 801–806, https://doi.org/10.1093/aob/mcg205, 2003.

Berkowitz, J. F., Page, S., and Noble, C. V.: Potential Disconnect between Observations of Hydrophytic Vegetation, Wetland Hydrology Indicators, and Hydric Soils in Unique Pitcher Plant Bog Habitats of the Southern Gulf Coast, Southeastern Naturalist, 13, 721, https://doi.org/10.1656/058.013.0410, 2014.

Bethke, C. M., Sanford, R. A., Kirk, M. F., Jin, Q., and Flynn, T. M.: The thermodynamic ladder in geomicrobiology, American Journal of Science, 311, 183–210, https://doi.org/10.2475/03.2011.01, 2011.

Blagodatsky, S., Blagodatskaya, E., Yuyukina, T., and Kuzyakov, Y.: Model of apparent and real priming effects: Linking microbial activity with soil organic matter decomposition, Soil Biology and Biochemistry, 42, 1275–1283, https://doi.org/10.1016/j.soilbio.2010.04.005, 2010.

Bräuer, S. L., Yavitt, J. B., and Zinder, S. H.: Methanogenesis in McLean Bog, an Acidic Peat Bog in Upstate New York: Stimulation by $H_2/CO_2$ in the Presence of Rifampicin, or by Low Concentrations of Acetate, 21, 433–443, https://doi.org/10.1080/01490450490505400, 2004.

Bridgham, S. D., Cadillo-Quiroz, H., Keller, J. K., and Zhuang, Q.: Methane emissions from wetlands: biogeochemical, microbial, and modeling perspectives from local to global scales, 19, 1325–1346, https://doi.org/10.1111/gcb.12131, 2013.

Catrouillet, C., Davranche, M., Dia, A., Bouhnik-Le Coz, M., Marsac, R., Pourret, O., and Gruau, G.: Geochemical modeling of Fe(II) binding to humic and fulvic acids, Chemical Geology, 372, 109–118, https://doi.org/10.1016/j.chemgeo.2014.02.019, 2014.

Chen, X., Xu, Y., Gao, H., Mao, J., Chu, W., and Thompson, M. L.: Biochemical stabilization of soil organic matter in straw-amended, anaerobic and aerobic soils, Science of The Total Environment, 625, 1065–1073, https://doi.org/10.1016/j.scitotenv.2017.12.293, 2018.

Chowdhury, T. R. and Dick, R. P.: Ecology of aerobic methanotrophs in controlling methane fluxes from wetlands, 65, 8–22, https://doi.org/10.1016/j.apsoil.2012.12.014, 2013.

Dahl, T. E.: Status and Trends of Wetlands in the Conterminous United States 2004 to 2009, 2011.

Darrah, S. E., Shennan-Farpón, Y., Loh, J., Davidson, N. C., Finlayson, C. M., Gardner, R. C., and Walpole, M. J.: Improvements to the Wetland Extent Trends (WET) index as a tool for monitoring natural and human-made wetlands, Ecological Indicators, 99, 294–298, https://doi.org/10.1016/j.ecolind.2018.12.032, 2019.

Deng, H., Ye, Z. H., and Wong, M. H.: Lead, zinc and iron (Fe2+) tolerances in wetland plants and relation to root anatomy and spatial pattern of ROL, Environmental and Experimental Botany, 65, 353–362, https://doi.org/10.1016/j.envexpbot.2008.10.005, 2009.

Drake, H. L., Horn, M. A., and Wüst, P. K.: Intermediary ecosystem metabolism as a main driver of methanogenesis in acidic wetland soil, 1, 307–318, https://doi.org/10.1111/j.1758-2229.2009.00050.x, 2009.

Ferry, J. G.: Methanosarcina acetivorans: A Model for Mechanistic Understanding of Aceticlastic and Reverse Methanogenesis, Frontiers in Microbiology, 11, 1806, https://doi.org/10.3389/fmicb.2020.01806, 2020.

Glissmann, K. and Conrad, R.: Saccharolytic activity and its role as a limiting step in methane formation during the anaerobic degradation of rice straw in rice paddy soil, Biology and Fertility of Soils, 35, 62–67, https://doi.org/10.1007/s00374-002-0442-z, 2002.

Gray, A. L.: Redoximorphic Features Induced by Organic Amendments and Simulated Wetland Hydrology, Master's Thesis, Univrersity of Maryland, 2010.

Green, H. E.: Use of theoretical and conceptual frameworks in qualitative research, 21, 34–38, 2014.

Guo, X., Liu, H., and Wu, S.: Humic substances developed during organic waste composting: Formation mechanisms, structural properties, and agronomic functions, Science of The Total Environment, 662, 501–510, https://doi.org/10.1016/j.scitotenv.2019.01.137, 2019.

Hansel, C. M., Lentini, C. J., Tang, Y., Johnston, D. T., Wankel, S. D., and Jardine, P. M.: Dominance of sulfur-fueled iron oxide reduction in low-sulfate freshwater sediments, 9, 2400–2412, https://doi.org/10.1038/ismej.2015.50, 2015.

Hondula, K. L., Jones, C. N., and Palmer, M. A.: Effects of seasonal inundation on methane fluxes from forested freshwater wetlands, Environ. Res. Lett., 16, 084016, https://doi.org/10.1088/1748-9326/ac1193, 2021.

Hu, Z., Liu, Y., Chen, G., Gui, X., Chen, T., and Zhan, X.: Characterization of organic matter degradation during composting of manure–straw mixtures spiked with tetracyclines, Bioresource Technology, 102, 7329–7334, https://doi.org/10.1016/j.biortech.2011.05.003, 2011.

Huang, B., Yu, K., and Gambrell, R. P.: Effects of ferric iron reduction and regeneration on nitrous oxide and methane emissions in a rice soil, Chemosphere, 74, 481–486, https://doi.org/10.1016/j.chemosphere.2008.10.015, 2009.

Hubbe, M. A., Nazhad, M., and Sánchez, C.: Composting as a Way to Convert Cellulosic Biomass and Organic Waste into High-Value Soil Amendments: A Review, 5, 47, 2010.

Jabłoński, S., Rodowicz, P., and Łukaszewicz, M.: Methanogenic archaea database containing physiological and biochemical characteristics, 65, 1360–1368, https://doi.org/10.1099/ijs.0.000065, 2015.

Keiluweit, M., Wanzek, T., Kleber, M., Nico, P., and Fendorf, S.: Anaerobic microsites have an unaccounted role in soil carbon stabilization, Nat Commun, 8, 1771, https://doi.org/10.1038/s41467-017-01406-6, 2017.

Kim, S., Lee, J., Lim, J., Shinog, Y., Lee, C., and Oh, T.: Comparison of Methane Emissions on Soil Texture in Korean Paddy Fields, 63, 393–397, https://doi.org/10.5109/1955660, 2018.

Kirk, G.: The Biogeochemistry of Submerged Soils, 1st ed., Wiley, https://doi.org/10.1002/047086303X, 2004.

Kuehn, K. A., Ohsowski, B. M., Francoeur, S. N., and Neely, R. K.: Contributions of fungi to carbon flow and nutrient cycling from standing dead Typha angustifolia leaf litter in a temperate freshwater marsh, Limnol. Oceanogr., 56, 529–539, https://doi.org/10.4319/lo.2011.56.2.0529, 2011.

Lalonde, K., Mucci, A., Ouellet, A., and Gélinas, Y.: Preservation of organic matter in sediments promoted by iron, 483, 198–200, https://doi.org/10.1038/nature10855, 2012.

Lehmann, J. and Kleber, M.: The contentious nature of soil organic matter, https://doi.org/10.1038/nature16069, 2015.

Lentini, C. J., Wankel, S. D., and Hansel, C. M.: Enriched iron (III)-reducing bacterial communities are shaped by carbon substrate and iron oxide mineralogy, 3, 2012.

Leprince, F. and Quiquampoix, H.: Extracellular enzyme activity in soil: effect of pH and ionic strength on the interaction with montmorillonite of two acid phosphatases secreted by the ectomycorrhizal fungus Hebeloma cylindrosporum, 47, 511–522, https://doi.org/10.1111/j.1365-2389.1996.tb01851.x, 1996.

Maietta, C. E., Hondula, K. L., Jones, C. N., and Palmer, M. A.: Hydrological Conditions Influence Soil and Methane-Cycling Microbial Populations in Seasonally Saturated Wetlands, Front. Environ. Sci., 8, 593942, https://doi.org/10.3389/fenvs.2020.593942, 2020a.

Maietta, C. E., Monsaint-Queeney, V., Wood, L., Baldwin, A. H., and Yarwood, S. A.: Plant litter amendments in restored wetland soils altered microbial communities more than clay additions, Soil Biology and Biochemistry, 147, 107846, https://doi.org/10.1016/j.soilbio.2020.107846, 2020b.

Marquart, K. A., Haller, B. R., Paper, J. M., Flynn, T. M., Boyanov, M. I., Shodunke, G., Gura, C., Jin, Q., and Kirk, M. F.: Influence of pH on the balance between methanogenesis and iron reduction, Geobiology, 17, 185–198, https://doi.org/10.1111/gbi.12320, 2019.

Mazzilli, S. R., Kemanian, A. R., Ernst, O. R., Jackson, R. B., and Piñeiro, G.: Greater humification of belowground than aboveground biomass carbon into particulate soil organic matter in no-till corn and soybean crops, Soil Biology and Biochemistry, 85, 22–30, https://doi.org/10.1016/j.soilbio.2015.02.014, 2015.

McMahon, S. K., Williams, M. A., Bottomley, P. J., and Myrold, D. D.: Dynamics of Microbial Communities during Decomposition of Carbon-13 Labeled Ryegrass Fractions in Soil, 69, 1238, https://doi.org/10.2136/sssaj2004.0289, 2005.

Megonigal, J. P., Hines, M. E., and Visscher, P. T.: Anaerobic Metabolism: Linkages to Trace Gases and Aerobic Processes, in: Biogeochemistry, Elsevier-Pergamon, Oxford, UK., 317–424, 2004.

Mikutta, R., Kleber, M., Torn, M. S., and Jahn, R.: Stabilization of Soil Organic Matter: Association with Minerals or Chemical Recalcitrance?, 77, 25–56, https://doi.org/10.1007/s10533-005-0712-6, 2006.

Mills, A. L., Herman, J. S., Hornberger, G. M., and DeJesús, T. H.: Effect of Solution Ionic Strength and Iron Coatings on Mineral Grains on the Sorption of Bacterial Cells to Quartz Sand, 60, 3300–3306, https://doi.org/10.1128/AEM.60.9.3300-3306.1994, 1994.

Mitsch, W. J., Bernal, B., and Hernandez, M. E.: Ecosystem services of wetlands, 11, 1–4, https://doi.org/10.1080/21513732.2015.1006250, 2015.

Muggeo, V. M. R.: segmented: An R Package to Fit Regression Models with Broken-Line Relationships, 8, 7, 2008.

Müller, C., Kuki, K. N., Pinheiro, D. T., de Souza, L. R., Silva, A. I. S., Loureiro, M. E., Oliva, M. A., and Almeida, A. M.: Differential physiological responses in rice upon exposure to excess distinct iron forms, 16, 2015.

Neubauer, S. C.: On the challenges of modeling the net radiative forcing of wetlands: reconsidering Mitsch et al. 2013, Landscape Ecol, 29, 571–577, https://doi.org/10.1007/s10980-014-9986-1, 2014.

Neubauer, S. C. and Verhoeven, J. T. A.: Wetland effects on global climate: mechanisms, impacts, and management recommendations, in: Wetlands: ecosystem services, restoration and wise use, Springer, 39–62, 2019.

Noble, R. and Roberts, S. J.: Eradication of plant pathogens and nematodes during composting: a review, 53, 548–568, https://doi.org/10.1111/j.0032-0862.2004.01059.x, 2004.

Nottingham, A. T., Griffiths, H., Chamberlain, P. M., Stott, A. W., and Tanner, E. V. J.: Soil priming by sugar and leaf-litter substrates: A link to microbial groups, Applied Soil Ecology, 42, 183–190, https://doi.org/10.1016/j.apsoil.2009.03.003, 2009.

Ott, E. T., Galbraith, J. M., Daniels, W. L., and Aust, W. M.: Effects of amendments and microtopography on created tidal freshwater wetland soil morphology and carbon, Soil Sci. Soc. Am. j., 84, 638–652, https://doi.org/10.1002/saj2.20057, 2020.

Paustian, K., Lehmann, J., Ogle, S., Reay, D., Robertson, G. P., and Smith, P.: Climate-smart soils, Nature, 532, 49–57, https://doi.org/10.1038/nature17174, 2016.

Pereira, E. G., Oliva, M. A., Siqueira-Silva, A. I., Rosado-Souza, L., Pinheiro, D. T., and Almeida, A. M.: Tropical Rice Cultivars from Lowland and Upland Cropping Systems Differ in Iron Plaque Formation, Journal of Plant Nutrition, 37, 1373–1394, https://doi.org/10.1080/01904167.2014.888744, 2014.

Poggenburg, C., Mikutta, R., Schippers, A., Dohrmann, R., and Guggenberger, G.: Impact of natural organic matter coatings on the microbial reduction of iron oxides, 224, 223–248, https://doi.org/10.1016/j.gca.2018.01.004, 2018.

Poulton, S. W. and Canfield, D. E.: Development of a sequential extraction procedure for iron: implications for iron partitioning in continentally derived particulates, 214, 209–221, https://doi.org/10.1016/j.chemgeo.2004.09.003, 2005.

Prakash, D., Chauhan, S. S., and Ferry, J. G.: Life on the thermodynamic edge: Respiratory growth of an acetotrophic methanogen, Sci. Adv., 5, eaaw9059, https://doi.org/10.1126/sciadv.aaw9059, 2019.

Richardson, C. J., Bruland, G. L., Hanchey, M. F., and Sutton-Grier, A. E.: Soil Restoration: The Foundation of Successful Wetland Reestablishment, in: Soil Restoration: The Foundation of Successful Wetland Reestablishment, vol. Chapter 19, CRC Press, 469–493, 2016.

Saaltink, R. M., Dekker, S. C., Eppinga, M. B., Griffioen, J., and Wassen, M. J.: Plant-specific effects of iron-toxicity in wetlands, Plant Soil, 416, 83–96, https://doi.org/10.1007/s11104-017-3190-4, 2017.

Saidpullicino, D., Erriquens, F., and Gigliotti, G.: Changes in the chemical characteristics of water-extractable organic matter during composting and their influence on compost stability and maturity, Bioresource Technology, 98, 1822–1831, https://doi.org/10.1016/j.biortech.2006.06.018, 2007.

Scott, B., Baldwin, A. H., Ballantine, K., Palmer, M., and Yarwood, S.: The role of organic amendments in wetland restorations, Restor Ecol, 28, 776–784, https://doi.org/10.1111/rec.13179, 2020.

Shimizu, M., Zhou, J., Schröder, C., Obst, M., Kappler, A., and Borch, T.: Dissimilatory Reduction and Transformation of Ferrihydrite-Humic Acid Coprecipitates, 47, 13375–13384, https://doi.org/10.1021/es402812j, 2013.

Sivan, O., Shusta, S. S., and Valentine, D. L.: Methanogens rapidly transition from methane production to iron reduction, 14, 190–203, https://doi.org/10.1111/gbi.12172, 2016.

Souza, R.: Optimal drainage timing for mitigating methane emissions from rice paddy fields, 9, 2021.

Straub, K. L., Benz, M., and Schink, B.: Iron metabolism in anoxic environments at near neutral pH, 34, 181–186, https://doi.org/10.1111/j.1574-6941.2001.tb00768.x, 2001.

Wang, Z. P., DeLaune, R. D., Patrick, W. H., and Masscheleyn, P. H.: Soil Redox and pH Effects on Methane Production in a Flooded Rice Soil, 57, 382, https://doi.org/10.2136/sssaj1993.03615995005700020016x, 1993.

Watanabe, K., Nishiuchi, S., Kulichikhin, K., and Nakazono, M.: Does suberin accumulation in plant roots contribute to waterlogging tolerance?, Front. Plant Sci., 4, https://doi.org/10.3389/fpls.2013.00178, 2013.

Willson, K. G., Perantoni, A. N., Berry, Z. C., Eicholtz, M. I., Tamukong, Y. B., Yarwood, S. A., and Baldwin, A. H.: Influences of reduced iron and magnesium on growth and photosynthetic performance of Phragmites australis subsp. americanus (North American common reed), Aquatic Botany, 137, 30–38, https://doi.org/10.1016/j.aquabot.2016.11.005, 2017.

Winton, R. S. and Richardson, C. J.: The Effects of Organic Matter Amendments on Greenhouse Gas Emissions from a Mitigation Wetland in Virginia's Coastal Plain, Wetlands, 35, 969–979, https://doi.org/10.1007/s13157-015-0674-y, 2015.

Yagi, K. and Minami, K.: Effect of organic matter application on methane emission from some Japanese paddy fields, 36, 599–610, https://doi.org/10.1080/00380768.1990.10416797, 1990.

Yang, W. H., McNicol, G., Teh, Y. A., Estera-Molina, K., Wood, T. E., and Silver, W. L.: Evaluating the Classical Versus an Emerging Conceptual Model of Peatland Methane Dynamics: Peatland Methane Dynamics, 31, 1435–1453, https://doi.org/10.1002/2017GB005622, 2017.

Yao, H. and Conrad, R.: Thermodynamics of methane production in different rice paddy soils from China, the Philippines and Italy, 11, 1999.

Yarwood, S. A.: The role of wetland microorganisms in plant-litter decomposition and soil organic matter formation: a critical review, 94, https://doi.org/10.1093/femsec/fiy175, 2018.

Ye, R., Jin, Q., Bohannan, B., Keller, J. K., McAllister, S. A., and Bridgham, S. D.: pH controls over anaerobic carbon mineralization, the efficiency of methane production, and methanogenic pathways in peatlands across an ombrotrophic–minerotrophic gradient, 54, 36–47, https://doi.org/10.1016/j.soilbio.2012.05.015, 2012.

Zhuang, L., Xu, J., Tang, J., and Zhou, S.: Effect of ferrihydrite biomineralization on methanogenesis in an anaerobic incubation from paddy soil: Iron Mineralization and Methanogenesis, J. Geophys. Res. Biogeosci., 120, 876–886, https://doi.org/10.1002/2014JG002893, 2015.