# Peer review of "Quantification of potential methane emissions associated with"

_Biogeosciences, 2021_

## Author Response (AR1)

**bg-2021-182**

Responses to comments:

Note to Editor: We have modified the references to match the journal's specifications, which were incorrect in the previous submission.

General response.

It is apparent from the comments that the scope and inference of the paper was not clear. To address this, we made several modifications. First, we modified the title to:

Quantification of potential methane emissions associated with organic matter amendments following oxic soil inundation

This also addresses a remark Commentor #1 about the title.

Also, one of the main points of the paper is the segmented gas production pattern, and there is a marked increase after the breakpoint. This was the case even without amendments, so inundation duration could be a means of controlling methane, which we will emphasize further. Some other comments wanted to steer the manuscript in a direction other than what we intended. For example, using  $CO_2$ :  $CH_4$  ratios to estimate carbon storage potential. We felt that would be inappropriate in this setting since plant root and methanotrophic activities are both missing. In these cases, we included the requested data, but maintained the manuscript's intended emphases.

Commentor #1

In this manuscript, the authors conduct microcosm experiments with two wetland soils, a sandy loam and a sandy clay loam, to explore how different organic amendments (from fresh to cured organic matter) affect CH4 emissions and Fe reduction. The paper addresses a topic of great interest to the biogeochemistry community, especially to those interested in mitigation efforts in wetlands. Amending soils with organic matter to increase soil carbon stocks is generally considered a key mitigation practice, so exploring systematically how different amendments affect emissions is important. The paper is also easy to follow, especially as it adopts a very simple structure. Overall, I believe the paper can be an important contribution, and I recommend publication after addressing some points of concerns described below. Generally, I think these points can be addressed by expanding the discussion and/or elaborating more on the methodology.

1. There needs to be more connection between the experiments being done and the type of wetland (and location within the wetland), for which the results are relevant. For instance, the experiments are conducted under anaerobic conditions, but this is not always the case in wetland soils, as some soils are affected by tidal fluctuations or are not necessarily inundated (e.g., peatlands). In non-inundated wetlands or seasonal wetlands, there might be an interplay between methanogens/methanotrophs and between different metabolic pathways to decompose carbon. So, it seems that the experiments are more relevant to

saturated/inundated wetlands (e.g., marshes or small lakes). I think that it would be important to read the authors' perspective on this.

The commenter makes makes several good points with respect to scope of inference, which we will need to clarify. First, although we tested two soils, both were sandy Atlantic coastal plain soils. We have discussed sandy soils specifically, and in our case the soil with the higher sand content produced more methane with OM amendments. However, there are numerous examples where this is not the case, which we have now cited.

The wetland type (e.g. "marsh" and "peatland") and location is one way to describe the relevant context. We believe the important factor is the duration of inundation after a dry period, which could happen in many contexts and is a more useful generalization. We added a reference to Drake et al., 2009, who reported a similar methane gas expression pattern in a peat soil, much different from our mineral soils.

Our experiment specifically excludes the activity of methanotrophs and we are only considering a continued inundation condition. Commenter #2 also also pointed this out. A better context to comment on this effect is a field study. We do have a companion field study, to be published, that shows the general pattern of methane production is predicted by this lab study, but the flux is much lower, likely due at least in part to the activity of methanotrophs. Also, we will address carbon storage in the field study, which includes the critical element of plant contributions.

2. Another important point is related to the overall conclusion of the study. That CH4 and CO2 emissions generally increase upon organic matter addition is expected, I would say. But how much do emissions increase relative to the amount of C provided? The authors should consider studying the emissions normalized by the amount of C added. This normalized measure could also be more relevant in the context of wetland management and restoration. Overall, if we add organic matter to wetland soils, we should expect an increase in emissions. But the questions are: how much of this organic matter ends up being emitted as CH4? How much as CO2? And how much will it be converted into stable organic carbon? Isn't it this partitioning that ultimately helps us decide whether adding a specific organic matter (and how much) is an effective mitigation/restoration or not?

We agree with the commenter and hypothesized that both  $CO_2$  and  $CH_4$  would increase with OM amendments (an expected outcome). This paper quantifies the increase by amendment type and dose, which we will emphasize further. In this way our results generally fit an expected gas production pattern, which we agree was not a new finding. However, by looking at this issue the way we did we are able to comment on other (sometimes more important) factors, which include inundation duration, pH, aeroturbation, and whether OM was composted.

We agree with the commentor that it is valuable to present methane emissions normalized by the amount of C added. This data is now included as Supplemental Figure S4. However, our

experiment resulted in few statistically significant differences so we felt we could not report that as a primary finding.

With respect to partitioning of OM into soil, using our  $CO_2$  numbers would be inappropriate since our system does not include photosynthetic  $CO_2$  loss and  $CO_2$ -producing methanotrophy. To properly account for total  $CO_2$  and  $CH_4$  emitted in a field context one would need time-averaged values, taking into account daily fluctuations over time, with a device such as an eddy covariance flux tower. Our companion field study, to be published separately, will be a better platform to address C partitioning as it can include soil inputs from plants.

3. There is a lot of material in the supplementary information, which could be included in the manuscript. Right now, as soon as one starts reading the results, one needs to stop and look for the supplementary Figures and Tables to be able to follow. If they are important for understanding the analysis and findings, they should be included in the main manuscript.

In addition to adding CO2:CH4 ratios, we moved the following supplemental figures into the main text: S4a (SCL vs SL, Figure 2), S5 (Typical gas production curve, Figure 1), and an example from S6 (Figure 4),

4. The observation that sandy loam has higher emissions than a sandy clay loam might seem trivial, if it is not discussed in more depth. Because of the higher clay content, I would assume that this is due to the higher specific surface area that tends to retain more carbon. If this is the reason, then this is well known. If there is more, then why do you suggest that they sandy loams are more vulnerable here? For example, in lines 235-239.

We have added Figure 2 along with additional discussion. In our study SL incubations emitted more methane than SCL. We cite other studies where this is also true, but cite still others where it was not. Overall, we felt qualifying this finding was the best option.

5. The implications of the study seem important, but the authors could elaborate more on them. I suggest the authors discuss more the implications, perhaps with some rough numbers estimated from their analysis. For example, the authors mention the design of systems that regulate flooding depending on the breakthrough time. I am surprised the authors are not mentioning/citing work on rice cultivations, where this technique of managing inundation to reduce emissions is widespread. In this regard, it seems that using organic amendment with long breakthrough times can be very important in rice fields. However, in rice fields, farmers tend to use rice straw as amendment, because of course it is readily available. What would be the implications for other wetland systems? Going back to point 1, linking the analysis to wetland type can be an important point of improvement.

This is an excellent point. We could have pulled more from rice cultivation studies. Fortunately, in the intervening time from submission, there has been a key publication, Souza 2021, which mirrors the findings of this lab study in a field setting. This reference will be included and discussed, along with several other relevant publications.

**Minor comments**

In the introduction, the different paragraphs are not well connected with each other. There is background material without explicit link to the overarching question. I suggest reframing a bit the introduction so that the research question is clear and so is the link to the background material in the various paragraphs.

Point taken. The abstract has a clearer flow, introducing methane emissions first then elaborating. We modified the introduction to follow the abstract flow and emphasized the theme of methane emissions.

The title mentions that adding organic material is not needed for hydric soil development, but this question is poorly discussed throughout the manuscript, so there seems to be a mismatch between title and manuscript. In my view, either the authors address this more explicitly in the manuscript, or they remove it from the title.

Agreed. We modified the title. Still, it is an important point because our results do not always follow the redox ladder paradigm. We have included several recent publications and extended our discussion to clarify this point.

Line 44: what do the authors mean by "couple it"?

We removed the phrase "couple it" and modified the text.

Fe reduction also depends on the amount of readily available Fe oxides and is not necessarily limited by available C. Did the authors consider this? Also, Fe reduction can be important in systems that experience oxic/anoxic fluctuations (or saturated/unsaturated conditions), because Fe reduction is very fast and if there is not an oxidation step where Fe2 is oxidized back to Fe3, then Fe reduction quickly stops. So, in what wetlands or wetland position do the authors think that this part of their analysis is important?

We saw, in several cases, that iron reduction did not "quickly stop", as the commenter points out, and continued even after methane generation increased (Figure 4). This was an unexpected finding. The fact that we observed a pattern, contrary to the commenter's point, is why we think this is important to include. With respect to redox fluctuations, many other studies address fluctuating redox conditions and we did not address that here, only the length of inundation following an oxic condition.

**Commentor #2**

The manuscript is focused on wetland soils affected by organic matter amendment and its effect on gases (carbon dioxide and methane) production and on iron reduction. The topic of the manuscript is important and actual. Carbon balance in wetland ecosystems, organic matter accumulation and greenhouse gases production are often studied within last ten years. In the study, two soils from newly constructed wetlands differing in soil texture and organic matter content were used and amended by four types of organic matter: hay, manure, compost, biosolid. Then gases production and iron reduction were studied under anaerobic conditions in the laboratory microcosm system.

In the result section I would expect to find the data which would show, what was the percentage of methane production from the total gas produced or the ratio of methane to CO2 production. Some data are shown in the supplemental table S2, but this table is not too helpful, as it is complicated to read it and to find the numbers, which I need. Some smaller clear summarizing table or graph of these data would be much better and it should be included in the manuscript.

**We added Table 1 which includes: $CO_2$ : $CH_4$ ratios and added some additional discussion that draws from $CO_2$ : $CH_4$ ratios.**

The authors did a conclusion that organic matter amendment to soil is dangerous as it increases methane emission from soil. However, methane emissions in the field were not measured but only potential methane production under laboratory conditions. It is therefore very difficult to make any conclusion about methane emissions from the soil in the field based only on data measured under controlled laboratory conditions. The problem is, that in the field many other factors affect methane emissions from the soil, which were not measured. These factors include especially water level, presence of vegetation and its composition, presence and activity of methane oxidizing bacteria (methanotrophs) in the aerobic layer on the soil surface and in the rhizosphere, presence and activity of anaerobic methane oxidizing bacteria in the soil, soil physico-chemical conditions like concentration of other electron acceptors in the soil profile (oxygen, nitrates, sulphates etc.).

We agree with both points: that methane emissions from field sites and consideration of other factors (e.g. methanotrophs) are important factors. This lab study is part of a much larger body of work that includes a field study, which we will publish separately. With respect to our choice to do a lab study in a jar, without plants and under nitrogen headspace to remove the activity of methanogens and plants - this was intentional. Our reported values represent methane production **potential**, not actual expected emissions. Our intention is to compare various OM amendments, and it is often the case in science that we do that by removing potentially confounding factors. This was discussed in the added reference Yang et al, 2016. This paper discusses the limitations of various approaches to estimating methane production and oxidation rates. As Yang points out, there is the potential for methanotrophy (including anerobic methane oxidation), so methane generation in soils may be underestimated. This work is a truer measure of potential methane production and has intentionally removed important factors such as methanotrophy and transport by plants. In that way it is a small but important part of a larger puzzle.

Methane production increased after addition of organic matter to soil, which is not surprising and it is known phenomenon. The data are then interpreted in sense that it is dangerous to add organic matter to soil due to increased methane production. But I miss any calculation, estimation or extrapolation of measured gases production rates to the field conditions. What happens in the field after organic matter amendment? How much would be methane emission from soil increased? And are then these values really much higher as compared to other sites, e.g. to natural wetlands? Is it possible to do such calculation/estimation based on the data measured under laboratory conditions? If we assume that constructed wetlands should function as natural wetlands, then there will be some methane emitted from flooded soil – this is nothing wrong, methanogenesis is natural process occuring in wetlands generally and it will never disseapear.

With respect, we disagree with the commenter on several points. We did not state OM amendments were dangerous. However, if we can learn a means to reduce methane emissions, we would consider that beneficial. Wetlands do emit methane naturally, but rather than accept this we are confident knowledgeable human action can limit emissions. We do agree that this is just one small piece of a larger picture and companion field studies are needed. We have performed a companion field study and will be reporting that the general patterns we observed in this lab study were evident in the field.

Also it is known, that wetlands are source of methane but it must be also taken into account that they accumulate carbon in soil organic matter. If these two processes are calculated and assumed in long-term (hundreds and thousands of years) the wetlands have generally cooling effect on both local and global climate because the effect of fixation of C to the soil is stronger than emission of methane to the atmosphere. This is also due to shorter retention time of methane in the atmosphere (so higher turnover rate) as compared to carbon dioxide. Moreover, there are other effects of organic matter amendment on soil characteristics and they may be even more important for soil and whole ecosystem functioning: effect on soil physico-chemical conditions like soil structure (soil aggregates, porosity, aeration), effect on humic substances, on sorption capacity (fixation of ions on humic substances), support of microbial activity and support of plant germination and growth etc. Generally, addition of organic matter to the soil and increase of its content in soil have positive effect on soil and it is desirable.

It is the lead author's opinion that scientific studies do not support the assertion that OM amendments are particularly beneficial for wetland restoration. See Scott et al., The role of organic amendments in wetland restorations, Restoration Ecology, 2020. While we would invite debate on that issue, we specifically did not mention the broader impacts of OM amendments because this study targets methane generation potential and some factors that control it. As an example, we have included Souza et al., Optimal drainage timing for mitigating methane emissions from rice paddy fields, Geoderma, 2021. They recommend, in spite of perceived benefits, discontinuing the use of rice straw amendments, or altering hydrology, to reduce excess methane emissions.

Therefore the conclusion.

We assume more was intended here. Since the comment period is closed, we appreciate your input and would welcome an email about this final comment and attempt to revise the manuscript accordingly.

The units used for gases production: "cc" – for me unusual, I guess, these are cubic centimeters? You should use either "mL" or "cm3" as SI units instead of "cc".

We will make this make this modification, subject to the journal's editorial requirements.

---

## Author Response (AR2)

1. The authors mentioned that this work is part of a larger project with a broader scope, which also includes field work. I am looking forward to future results from the project. Perhaps it might be of interest to the readership to read in the manuscript that this manuscript presents the first of a series of contributions from a broader project. I would leave it up to the authors to decide whether to mention this in the introduction or maybe in the conclusions.

*This reference is included on line 438.*

2. The authors could add a table that summarizes the details of all the experiments. This could be Table 1.

*A Table 1 is now included for the editor's consideration.*

3. Within section 3, but before subsection 3.1, it would be helpful to include a summary of how the results will be presented in later sections. I believe it would make the reading smoother.

*We added a short paragraph to Section 3.0 that references the new summary Table 1 (per comment #2).*

I also found a couple of typos while reading the manuscript.

Line 73. Replace "iron" with "Fe". Fe was introduced in line 72.

*It was our understanding that it is inappropriate to begin a sentence with an abbreviation.*

Line 225. "were fitted with".

*Changed as requested.*

*Note to editor: Hyperlinks, which we use to aid in the editing process, have been removed and appear in tracked changes.*